# *FZD7*, Regulated by Non-CpG Methylation, Plays an Important Role in Immature Porcine Sertoli Cell Proliferation

**DOI:** 10.3390/ijms24076179

**Published:** 2023-03-24

**Authors:** Anqi Yang, Saina Yan, Yanfei Yin, Chujie Chen, Xiangwei Tang, Maoliang Ran, Bin Chen

**Affiliations:** College of Animal Science and Technology, Hunan Provincial Key Laboratory for Genetic Improvement of Domestic Animal, Hunan Agricultural University, Changsha 410128, China

**Keywords:** *FZD7*, non-CpG methylation, Sertoli cells, testes, proliferation, apoptosis

## Abstract

The regulatory role of non-CpG methylation in mammals has been important in whole-genome bisulfite sequencing. It has also been suggested that non-CpG methylation regulates gene expression to affect the development and health of mammals. However, the dynamic regulatory mechanisms of genome-wide, non-CpG methylation during testicular development still require intensive study. In this study, we analyzed the dataset from the whole-genome bisulfite sequencing (WGBS) and the RNA-seq of precocious porcine testicular tissues across two developmental stages (1 and 75 days old) in order to explore the regulatory roles of non-CpG methylation. Our results showed that genes regulated by non-CpG methylation affect the development of testes in multiple pathways. Furthermore, several hub genes that are regulated by non-CpG methylation during testicular development—such as *VEGFA*, *PECAM1*, and *FZD7*—were also identified. We also found that the relative expression of *FZD7* was downregulated by the zebularine-induced demethylation of the first exon of *FZD7*. This regulatory relationship was consistent with the results of the WGBS and RNA-seq analysis. The immature porcine Sertoli cells were transfected with RNAi to mimic the expression patterns of *FZD7* during testicular development. The results of the simulation test showed that cell proliferation was significantly impeded and that cell cycle arrest at the G2 phase was caused by the siRNA-induced *FZD7* inhibition. We also found that the percentage of early apoptotic Sertoli cells was decreased by transfecting them with the RNAi for *FZD7*. This indicates that *FZD7* is an important factor in linking the proliferation and apoptosis of Sertoli cells. We further demonstrated that Sertoli cells that were treated with the medium collected from apoptotic cells could stimulate proliferation. These findings will contribute to the exploration of the regulatory mechanisms of non-CpG methylation in testicular development and of the relationship between the proliferation and apoptosis of normal somatic cells.

## 1. Introduction

Sertoli cells are non-uniformly shaped and are the biggest cells in testes. They are located in the germinal epithelium, in the seminiferous tubules [1]. One of the most important features of Sertoli cells is that they secrete the Mullerian inhibiting factor, which prevents the development of female sex organs after the testes are determined embryologically [2]. Sertoli cells can also promote the formation of the blood–testis barrier and spermatogenesis by secreting substances such as inhibin B, the androgen binding protein (ABP), and testicular fluid [3]. Notably, approximately 30–50 germ cells are supported by a single Sertoli cell during the different developmental stages [4], and the population of Sertoli cells is relatively stable after testicular maturation [5]. A previous study suggested that Sertoli cells in mature testes are terminally differentiated, and that the number of germ cells, the size of a testis, and the sperm output of an adult testis depends on the number of Sertoli cells [6]. Therefore, it can be considered that the proliferation of immature Sertoli cells directly relates to the function of adult testes.

DNA methylation (the presence of 5-methylcytosine, 5mC) is an epigenetic marker and regulates gene expression without altering the genomic sequence. Previous investigations have shown that DNA methylation plays key regulatory roles in cell differentiation, normal development, genomic imprinting, X-chromosome inactivation, and the suppression of parasitic DNA sequences [7]. In mammals, approximately 60–80% of CpG dinucleotides are methylated; however, non-CpG methylation has a relatively high proportion in some special cells, such as oocytes, embryonic stem cells (ESCs), and nerve cells [8]. Non-CpG was identified as a novel epigenetic modification in the early 1980s [9]; however, the potential functional roles of non-CpG in developmental biology was first revealed after the 2000s [10]. The functional roles of non-CpG methylation in mammals at an early stage can be easily ignored because of the immaturity of sequencing technology. However, recent studies have shown that non-CpG methylation has multiple biological functions in different tissues. For example, the increase in the non-CpG methylation level in the peroxisome proliferator-activated receptor-gamma coactivator (*PGC-1α*) promoter was associated with impaired glucose tolerance in a type 2 diabetes model [11]. In addition, the expression of *Presenilin 1* was negatively regulated by the non-CpG methylation level associated with the promoter in Alzheimer’s disease (AD) samples [12]. It has also been reported that non-CpG methylation patterns dynamically change during testicular development [13,14]; therefore, the functions of non-CpG methylation on the proliferation and differentiation of testicular cells are worthy of study.

Our recent research showed that DNA methylation in CpG contexts that are associated with promoters and gene bodies affects the proliferation and differentiation of testicular cells by regulating the gene expression [15]. However, the regulatory mechanisms of non-CpG methylation during testicular development were not elucidated in our past research; therefore, in this study, we re-analyzed the data from the whole-genome bisulfite sequencing (WGBS) in order to identify the differentially methylated regions (DMRs) and the differentially methylated genes (DMGs) in non-CpG contexts, between the 1- and 75-day-old stages. The relationship that we observed between non-CpG methylation and gene expression suggested that non-CpG methylation affects testicular maturation through multiple pathways. We further showed that *FZD7*, when positively regulated by non-CpG methylation, promotes the proliferation of Sertoli cells by inducing early apoptosis. Together, our results provide valuable information and novel insights into the regulatory mechanisms of non-CpG methylation during testicular development.

## 2. Results

### 2.1. Characterization of DMRs in Non-CpG Contexts

In previous studies, we have reviewed testicular genomic methylation patterns at different developmental stages, and it is generally believed that the sequencing data for the samples have good repeatability at the global level of the methylation levels [15]. Therefore, in the present study, we directly performed a series of analyses on the potential regulatory mechanisms of non-CpG methylation during testicular maturation. A total of 1157 DMRs (hypermethylated DMRs: 590 + hypomethylated DMRs: 567) in the CHG (where H is T, C, or A) context and 2920 DMRs (hypermethylated DMRs: 1105 + hypomethylated DMRs: 1815) in the CHH context were identified (Appendix A). The length of the DMRs in both the CHG and CHH contexts ranged from 51 to 349 and from 51 to 2232 bp, respectively. These measurements did not conform to the normal etheric distribution (Figure 1A). The circos diagrams were drawn with points representing the significance of the differences among the DMRs. As shown in Figure 1B, there were a large number of DMRs in the genome during testicular development. To further compare the distribution of the DMRs in various functional genomic elements between birth and maturation and the methylation status of different regions, we analyzed the CGIs, CGI shores, promoters, TSSs, 5′UTRs (5-untranslated regions), exons, introns, 3′UTRs (3-untranslated regions), repeats, and other regions. We found that the number of DMRs in non-CpG contexts was the highest in repeat, followed by introns, and in other regions these were generally less (Figure 1C). In addition, as shown in the heat maps in Figure 1D, hierarchical clustering was used to analyze the methylation levels of DMRs at the stages of birth and maturation, and there was a clear separation between the two developmental stages. These results showed the dynamics of non-CpG methylation during testes’ development.

### 2.2. Identification of DMGs in Non-CpG Contexts

In the present study, DMGs were identified as DMR-related genes when they overlapped with the gene bodies (from TSS to TSE) or promoters (Appendix A). A total of 600 DMGs in CHG contexts were detected, among which 570 DMGs were identified as DMRs in gene bodies—including 301 hypermethylated DMGs and 291 hypomethylated DMGs—and 22 DMGs were shared in the hypermethylated and hypomethylated DMGs associated with gene bodies. In addition, 45 DMGs in CHG contexts were identified as DMRs in promoters, including 14 hypermethylated DMGs and 31 hypomethylated DMGs. A total of 15 DMGs were shared among those DMGs associated with gene bodies and promoters (Figure 1E).

Similarly, in the CHH context, a total of 1406 DMGs were detected. Among these, 1329 DMGs were identified as DMRs in gene bodies—including 563 hypermethylated and 840 hypomethylated DMGs—and 74 DMGs were shared in the hypermethylated and hypomethylated DMGs associated with gene bodies. In addition, in CHG contexts, 122 DMGs were identified as DMRs in promoters—including 37 hypermethylated and 85 hypomethylated DMGs—and ten DMGs were shared in the hypermethylated and hypomethylated DMGs associated with promoters. A total of 45 DMGs were shared among those DMGs that were associated with gene bodies and promoters (Figure 1F).

### 2.3. The Correlation between DNA Methylation and Gene Expression

At the chromosome level, the DNA methylation in non-CpG contexts showed a particular abundance of transposable elements (TEs) and gene expression (Figure 2A). Furthermore, all genes were divided into four groups (no-, low-, medium-, and high-expression) according to the RNA-seq data at each stage, and we showed the corresponding relationships between the DNA methylation level and the gene expression level near the gene bodies. As shown in Figure 2B, the hypomethylation in the promoter near the TSSs corresponded to the high gene expression at two developmental stages, while those in the gene bodies near the TSSs matched the low gene expression. To intuitively understand the relationship between non-CpG methylation and gene expression, the variations in the differential gene expression levels, which correspond to the differential methylation levels of the DMGs, are depicted in Figure 2C. As shown in Figure 2C, the effects of the differential methylation levels on DMG expression were random, and this phenomenon can be explained by the fact that DNA methylation is only one of the several regulatory mechanisms of gene expression. However, the results from the Pearson correlation analysis showed that the differential methylation levels correlated negatively with the differential expression of the DMGs associated with promoters (r = −0.0533), whereas the differential methylation levels correlated positively with the differential gene expression of the DMGs associated with gene bodies (r = 0.0274) (Appendix A). In addition, previous investigations have revealed that both CpG and non-CpG methylation were based on the same enzymes [16,17], and they were usually coincident in the genome [18]. Therefore, the overlapping DMGs associated with both CpG and non-CpG may have interfered with our conclusions about the function of non-CpG methylation. As shown in Figure 2D, a total of 726 overlapping DMGs, related to both CpG and non-CpG contexts, were detected. These genes were rejected in the next analysis. To reduce the impact of other regulatory mechanisms on our conclusions, we screened out the overlapping genes between the DMGs associated with the promoters and DEGs, in which the differential methylation levels regulated negatively with the differential expression of the corresponding genes. These overlapping genes were identified as PNGs. Similarly, we also screened out the overlapping genes between the DMGs associated with gene bodies and DEGs, of which differential methylation levels regulated positively with the differential expression of corresponding genes. These genes were identified as BPGs.

### 2.4. Enrichment Analyses

A total of 338 BPGs and 47 PNGs were simultaneously used in the enrichment analyses. In the GO enrichment analysis, 136 significantly over-represented GO terms were identified for the regulation of the overlapping genes. The leading 20 GO terms, annotated at Level 2 for three main categories (molecular function, cellular component, and biological process), are presented in Figure 3A. For example, the leading GO terms in the three different categories were the single-multicellular organism process (GO:0044707), the cell projection (GO:0042995), and the single-multicellular organism process (GO:0044707)(Figure 3B). The KEGG analysis identified a total of 222 pathways (Figure 3C), of which several pathways were related to testicular cells’ proliferation, such as the PI3K-Akt signaling pathway (ssc04151) and the MAPK signaling pathway (ssc04010) in testicular cells [19]. The results from the enrichment analysis indicated that gene expressions regulated by non-CpG methylation were associated with testicular development and maturation. In addition, to verify the reliability of the RNA-seq data, three randomly selected PNGs (*IFITM1*, *GAREM2*, and *TCEAL9*) and BPGs (*FZD7*, *RYK*, and *IL2RB*) from the testes samples, taken at two developmental stages, were used for qPCR. As shown in Figure 3D, the results from the q-PCR assay were generally consistent with the data generated in the RNA-seq.

### 2.5. Screening of Hub Genes Regulated by Non-CpG Methylation

To construct the protein–protein interaction (PPI) network, the aforementioned BPGs and PNGs were analyzed in the STRING online database (http://string-db.org (accessed on 13 December 2022) Version: 11.5, Geneva, Switzerland). After calculating the degree of the leading genes, the PPI networks were displayed using the CytoHubba plug-in within Cytoscape (Figure 3E). Next, these two functional modules from the PPI networks were screened using the MOCODE plug-in (Figure 3F). The analytical results from the CytoHubba plug-in and the MOCODE plug-in indicated that *VEGFA*, *PECAM1*, and *FZD7* were the primary three genes regulated by the non-CpG methylation during testicular maturation (Table 1).

### 2.6. Zebularine Treatment

Data from the WGBS and RNA-seq showed that the differential methylation levels of the DMGs associated with gene bodies and promoters were positively and negatively correlated with differential expression; however, in vitro assays are still necessary to avoid statistical errors. For this reason, the immature Sertoli cells were treated using zebularine (a specific inhibitor of DNA methyltransferase) and DMSO (negative control (NC)), and then we performed a bisulfite sequencing PCR (BSP) to analyze the changes in the methylation levels of the promoter in *IFITM1* and the first exon in *FZD7* (Figure 4A). The results from the BSP showed that the methylation levels in the promoter of *IFITM1* and the first exon of *FZD7* declined in the presence of zebularine, relative to DMSO (Figure 4B,C and Table 2). However, the results of the qPCR suggested that the expression of *FZD7* and *IFITM1* in zebularine-treated Sertoli cells were downregulated and upregulated, respectively (Figure 4D). These results revealed that dynamically methylated non-CpG is a novel regulatory mechanism of gene expression during testicular development. This was consistent with the results of the WGBS and RNA-seq.

### 2.7. FZD7 Promotes Sertoli Cell Proliferation

In our study, *FZD7*, a member of the frizzled family of Wnt signaling receptors, was the hub gene regulated by non-CpG methylation during testicular maturation and was expressed in the porcine Sertoli cells (Figure 5A). To mimic the *FZD7* expression in vivo and to explore the regulatory roles of *FZD7* in Sertoli cell proliferation, a specific siRNA was designed to knockdown the expression of the *FZD7* gene in immature porcine Sertoli cells (Figure 5B). The results of the cycle assay revealed that the *FZD7* interference decreased the percentage of cells in the G2 phase (Figure 5C,D). In addition, the relative expression of the genes related to the cell cycle, including *CDK4*, *CCND1*, and *CNNE1*, were detected using the qPCR. As shown in Figure 5E, *CCND1* and *CNNE1* were downregulated due to *FZD7* interference. Next, the CCK-8 assay showed that the cell proliferation index was reduced by the *FZD7* interference (*p* < 0.05) (Figure 5F). Similarly, the *FZD7* knockdown significantly decreased the relative mRNA expression of the cell proliferation-related genes, including *EGF*, *PCNA*, *GDNF*, *FGF*, *IGF*, and *BMP* (Figure 5G). Obviously, these results demonstrate that *FZD7* promotes immature porcine Sertoli cell proliferation.

### 2.8. FZD7 Increases Intracellular Ca^2+^ Concentrations and Adenosine Triphosphate (ATP) Levels

The results from the measurement of the Ca^2+^ concentrations showed that the intracellular Ca^2+^ levels decreased significantly due to *FZD7* interference (*p* < 0.05) (Figure 5H). In mammalian mitochondria, the increased supply of reducing equivalents—in the form of FADH2 and NADH—which is caused by the Ca^2+^ activation of the dehydrogenases promotes ATP synthesis. As expected, the ATP levels of the cells treated with siRNA significantly decreased (*p* < 0.05) (Figure 5I), suggesting that *FZD7* also promotes cell proliferation through the Wnt/Ca^2+^ signaling pathway.

### 2.9. FZD7 Induces Early Apoptosis of Sertoli Cells

We also examined the potential roles of *FZD7* in Sertoli cell apoptosis. The results of the flow cytometry suggested that the percentage of early apoptotic cells was decreased by the *FZD7* interference (*p* < 0.05); however, there was no significant difference in either the percentage of late apoptotic cells or the apoptosis rate between the two groups (*p* > 0.05) (Figure 6A,B). Next, western blotting analyses showed that the protein expression of BAX and Caspase-3, which both play a vital role in apoptosis, was decreased by the *FZD7* knockdown; however, the expression of Bcl2, an anti-apoptosis protein, was upregulated. The differential expression of these proteins seemed to be related to the blockage of the Wnt/β-catenin signaling pathway, caused by *FZD7* interference (Figure 6C). In summary, these data revealed that the *FZD7* gene promoted early apoptosis in Sertoli cells. Moreover, the results from the qPCR showed that the relative expression of *c-MYC* was blocked by the siRNA-induced *FZD7* inhibition (Figure 6D). Furthermore, we also observed that the levels of prostaglandin E2 (PGE_2_) were significantly decreased by the siRNA-induced *FZD7* inhibition (*p* < 0.05) (Figure 6E). These results indicate that *FZD7* is an important factor in linking the cell apoptosis and proliferation of immature Sertoli cells. We believe that the immature Sertoli cell repopulation, which is stimulated by cell apoptosis, also occurs during testicular development. To test our hypothesis, we performed the apoptosis medium treatment to determine whether the substances produced by dying, immature Sertoli cells could stimulate the growth of living cells. As shown in Figure 6F,G, the relative ATP level and the relative expression of several genes associated with proliferation increased when we treated them with an apoptosis medium for 48 h (*p* < 0.05). In addition, the results of the CCK-8 assay showed that the cell proliferation index was also increased when we treated them with an apoptosis medium for 72 h (*p* < 0.05) (Figure 6H).

## 3. Discussion

WGBS is the current ideal for DNA methylation analysis, as it provides high-resolution, single-based methylation maps [20], which enable non-CpG methylation in mammals to be fully understood. Previous studies have suggested that non-CpG methylation is more abundant in stem cells than in differentiated cell types. For example, 5mC at CpA sites accounts for approximately 25% of all 5mC in human embryonic stem cells, whereas the proportion of those in human fibroblast cell lines is negligible [21]. In the porcine testicular genome, non-CpG methylation levels averaged 1.06% [15]. However, recent studies have reported that a high proportion of non-CpG methylation is also present in differentiated cells, for example, brain tissue [22,23]. Obviously, reports describing the effects of non-CpG methylation on gene expression can provide more convincing evidence for its functional role in mammalian cells than the amount of non-CpG methylation in genomes can. The methylation of Cp^m^CpNpGpG sites within the *B29* gene promoter region has been proven to directly block the binding of early B-cell factor transcription factor and to reduce the activity of human B cells [24]. A study on epigenetic modification patterns in breast cancer have revealed that *HIF-1α* transcription was silenced by the non-CpG methylation in the promoter [25]. Numerous studies that have been conducted on mammals have prevented us from overlooking the roles of non-CpG methylation during testicular maturation [11,26,27]; therefore, we proposed that non-CpG methylation might play important regulatory roles during testicular development.

In the present study, a total of 4077 DMRs and 1819 DMGs in non-CpG contexts were identified—less than those in CpG contexts. According to our combined analysis results, overall, the non-CpG methylation levels in promoters and gene bodies were negatively and positively correlated with gene expression, respectively. This was similar to DNA methylation in CpG contexts [15], which indicates that DNA methylation in both CpG contexts and non-CpG contexts might own the same regulatory mechanism in gene expression. DNA methylation and demethylation can change the accessibility of the promoter and can affect the recruitment of DNA-binding proteins. For example, DNA methylation can repress the transcription activation of CGI promoters containing their sequence-recognition motifs [28]. Moreover, DNA methylation can promote heterochromatin formation by recruiting remodelers and modifiers to chromatin through DNA methyltransferase proteins [29]. However, the positive regulatory mechanism of DNA methylation in gene bodies has not been fully elucidated, although two hypotheses have been proposed to explain this phenomenon: (1) 5mC in a gene body facilitates co-transcriptional and/or splicing transcription elongation and (2) 5mC in a gene body inhibits intragenic promoters [28,30]. Based on the above-mentioned clues, CpG methylation and non-CpG methylation occur alternately in the genome, so we were unable to determine whether CpG methylation or non-CpG methylation is responsible for the differential expression of the overlapping DMGs associated with CpG and non-CpG contexts. Therefore, these overlapping DMGs were discarded for enrichment analysis and the PPI network construction. Undoubtedly, gene expression is regulated by various mechanisms during mammalian development, such as miRNA [19], lncRNA [31], and circular RNA [32]; therefore, not all of the differential expression levels of DMGs conform to the above-mentioned regulatory relationship. Consequently, this study mainly focused on BPGs and PNGs.

An enrichment analysis is an important way to identify the functional relationships of DMGs and to gain mechanistic insight into the genes regulated by non-CpG methylation. Enriched GO terms revealed that most of the genes regulated by non-CpG methylation were correlated with cell proliferation, cell differentiation, and basic development—including cellular processes, developmental processes, multicellular organismal processes, single-organism processes, cells, cell parts, and binding. The results of the GO analysis also indicated that non-CpG methylation tended to regulate gene expression during testicular maturation. KEGG pathway analyses showed that genes regulated by non-CpG methylation were mainly enriched in a series of important pathways related to testicular development and spermatogenesis, including steroid hormone biosynthesis (ssc00140) and linoleic acid metabolism (ssc00591). Testosterone, an important steroid hormone synthesized by Leydig cells, is essential in maintaining male characteristics and spermatogenesis [33]. Linoleic acid can promote testosterone synthesis by activating the GPR120/ERK pathway in Leydig cells [34]. Additionally, the primary three hub genes that were regulated by non-CpG methylation were identified as *VEGFA*, *PECAM1*, and *FZD7.* Sex-specific vascularization is a hallmark of testis differentiation, and it occurs as endothelial cells migrate from the adjacent mesonephros into a testis to surround the Sertoli germ cell, where they aggregate and guide seminiferous cord formation [35]. Previous research has shown that vascular endothelial growth factor A (*VEGFA*) is a key regulator during sex-specific vascularization processes [36,37]. The platelet endothelial cell adhesion molecule-1 (*PECAM1*) is a marker of endothelial and germ cells and is related to angiogenesis in the testes [38]. In summary, our results in this study suggest that non-CpG methylation affects testicular maturation in multiple ways.

The Wnt signaling pathway is an evolutionarily conserved signal transduction cascade, which relates to cell polarization, migration, and differentiation [39]. Three Wnt signaling pathways have been summarized in this paper: the canonical Wnt pathway (Wnt/β-catenin), the noncanonical Wnt/Ca^2+^ pathway, and the noncanonical Wnt/planar cell polarity (PCP) pathway. In addition, the Wnt signaling pathway has been shown to be involved in testis determination, and in the development of testicular somatic cells and germ cells [40]. For example, Wnt/β-catenin signaling was inhibited by the binding of SRY onto β-catenin, through which the formation of testicles begins [41]. Furthermore, it has been reported that several proteins involved in the intercellular tight junctions in bovines Sertoli cells and polarity maintenance are regulated by *WT1*, through the Wnt/PCP pathway. This has an important effect on the blood–testis barrier formation [42]. Moreover, the molecular significance of the Wnt/Ca^2+^ pathway is that it regulates gene expressions by affecting the intracellular calcium deposition [43]. Intracellular Ca^2+^ are the virtual regulators of all cellular processes, and those concentrations are affected by the Wnt/Ca^2+^ signaling pathway. A key step in the cell cycle has been reported to be Ca^2+^ signaling-dependent [44]. Moreover, the exit from quiescence in the early G1 phase is induced by an increased Ca^2+^ concentration [45]. However, cell proliferation cannot be promoted by increasing the Ca^2+^ concentration alone, but also requires the participation of several factors, such as *MYC* [46]. Additionally, several cysteine proteases and calpains require Ca^2+^ for their activation and are important mediators of apoptosis, indicating that cell apoptosis is regulated by Ca^2+^ [47]. It is noteworthy that the activation of all three pathways depends on the binding of the Wnt protein to the frizzled family receptors.

*FZD7* was identified as the most important hub gene, as it encodes the frizzled-7 (*FZD7*) protein, which is an important member of the frizzled family. However, the role of *FZD7* in Sertoli cell proliferation and apoptosis still requires intensive study, as the mechanisms of these have not been fully investigated. In this study, before conducting the in vitro simulation experiments, the Sertoli cells were treated using zebularine. The results of the BSP and qPCR showed that the expression of *FZD7* was positively regulated by non-CpG methylation, which rendered our results more convincing. Our results showed that *FZD7* could promote cell proliferation and early apoptosis. Interestingly, cell proliferation and apoptosis are not opposing events during development. For example, the slow-growing cells are cleared by the fast-growing cells—this competitive cellular interaction is called cell competition [48]. Moreover, previous studies have reported that cell apoptosis can induce proliferation, even though this phenomenon is mainly presented in damaged tissue or tumor cells [49,50]. In the present study, *c-MYC*, which both promotes cell apoptosis and stimulates cell proliferation [51], was downregulated by the *FZD7* interference, suggesting a link between the apoptosis and proliferation of immature Sertoli cells. It is undisputed that the proliferation of tumor cells is faster than apoptosis, and that normal somatic cells necessarily act in the same way during development. As shown in Figure 6B, the apoptosis rate of the immature Sertoli cells was not changed by the *FZD7* interference, whereas the percentage of early apoptosis cells decreased, which can be explained by the fact that the early stages of apoptosis are reversible [52]. Both *FZD7* and *c-MYC* are considered to be potential markers of tumor cells [53,54]; therefore, we speculated that the link between proliferation and apoptosis in the immature Sertoli cells was similar to that in tumor cells, and that this mechanism was influenced by the expression level of *FZD7*. A mechanism involved in cell repopulation, stimulated by cell apoptosis, named the “Phoenix Rising” pathway, has revealed that PGE_2_—an important regulator of cell proliferation—is positively regulated by the activation of Caspase-3 during cell apoptosis [50]. Notably, it has been reported that calcium independent phospholipase A2 (iPLA2) is activated in a Caspace-3 dependent manner during cell apoptosis—which promotes the release of arachidonic acid (AA)—and that PGE_2_ is a downstream product of AA [50,55,56]. Additionally, the Wnt signaling pathway is involved in apoptosis-stimulated tissue regeneration [57], and the Wnt signaling cascade at the level of β-catenin is modified by PGE_2_ through cAMP/PKA-mediated stabilizing phosphorylation events [58]. Therefore, we believe that *FZD7* could promote the proliferation of immature Sertoli cells by inducing early apoptosis (Figure 7). Overall, our results also suggested that the down-regulation of the *FZD7* expression by non-CpG demethylation may be one of several factors that maintains the relative stability of the size of mature testes. In addition, it is necessary to emphasize that the proteins related to cell apoptosis, which stimulate proliferation, may not be limited to PGE_2_. It is also necessary to emphasize that the mechanism of the *FZD7* expression, when it is positively regulated by non-CpG methylation in gene bodies, has not been fully clarified yet. Thus, we will improve further research studies by using new technologies.

In conclusion, our study highlighted the regulatory and functional roles of non-CpG methylation during normal testicular maturation and explored the potential role of *FZD7* in Sertoli cell proliferation. These results suggested that non-CpG methylation and CpG methylation had similar effects on gene expression in the testicular genome. Therefore, these findings could enhance our comprehensive understanding of the regulatory mechanisms of non-CpG methylation during testicular development.

## 4. Materials and Methods

### 4.1. Ethics Statement

All the experimental procedures in the present study were performed according to the guidelines of the Declaration of Helsinki. Sample collection was discussed and approved by the ethics committee of Animal Science and Technology College of Hunan Agricultural University (No. 2021–13). The boars did not suffer unnecessarily at any stage.

### 4.2. Data Set from WGBS and RNA-Seq

Testicular tissue samples were collected from Shaziling boars at two developmental stages: 1-day-old and 75-days-old (named SZ_1_d and SZ_75_d). Each group contained three biological replicates. Both WGBS and RNA-seq libraries were constructed on an Illumina Novaseq 6000, by Novogene Corporation (Beijing, China), and 150 bp paired-end reads were generated. An overview of the basic analysis of WGBS and RNA-seq were conducted in our past research, and differentially expressed genes (DEGs) in two developmental stages were also identified [15]. The raw sequence data of WGBS and RNA-seq obtained in our previous studies was deposited in the Genome Sequence Archive (Genomics, Proteomics and Bioinformatics, 2021), in National Genomics Data Center (Nucleic Acids Res 2022), China National Center for Bioinformation/Beijing Institute of Genomics, and Chinese Academy of Sciences (GSA: CRA009180 and CRA009195). These are publicly accessible at https://ngdc.cncb.ac.cn/ (accessed on 12 December 2022) [59,60].

### 4.3. Annotation of Gene and Genomic Features

The reference genome (*Sscrofa11.1*) was used to obtain gene annotation data. The region between the transcription start site (TSS) and the transcription termination site (TTS) was defined as the gene body region, and the 2-kb region, upstream of the TSS, was identified as the promoter region. CpG islands (CGIs) were defined as the CpG dinucleotides that tended to cluster in CpG contexts. CGIs were defined as regions > 200 bp, with a GC fraction > 0.5 and an observed-to-expected ratio of CpG > 0.65 on repeat-masked sequences, as annotated by the UCSC Genome Browser. The 2-kb regions extending in both directions from CGI were defined as CGI shore. The CGI and CGI shore were annotated using the UCSC Genome Browser [61].

### 4.4. Identification of DMRs and DMGs in Non-CpG Contexts

The DSS software was used to identify the DMRs by estimating the dispersion parameter from gamma-Poisson or beta-binomial distributions [62,63]. Next, genes whose gene body region (from TSS to TTS) or promoter region (2 kb upstream from the TSS) overlapped with the DMRs were identified as the DMGs.

### 4.5. Correlation Analysis of DNA Methylation and Gene Expression

To confirm the relationship between non-CpG methylation and gene expression, genes were divided into four groups, based on their expression (no-, low-, medium-, and high-expression), using quartile method, as follows: number of fragments per kilobase of exon per million fragments (FPKM) < 1; low, 1 ≤ FPKM < lower quartile; medium, lower quartile ≤ FPKM < upper quartile; and high, upper quartile ≤ FPKM. We observed that the DNA methylation levels of the gene body and the 2 kb region extended in both directions from the gene body in the four gene groups, at the two developmental stages. Pearson correlation coefficients between the differential methylation level of the promoter/gene body and the differential expression level of the corresponding gene were calculated in testes across the two developmental stages.

### 4.6. Enrichment Analysis

Gene ontology (GO) and Kyoto Encyclopedia of Genes and Genomes (KEGG) pathway enrichment analyses of BPGs and PNGs were performed using the OmicShare tools—a free online platform for data analysis (https://www.omicshare.com/tools (accessed on 18 December 2022)). Both GO terms and KEGG pathways with a *p*-value < 0.05 were considered nominally significant.

### 4.7. Construction and Module Analysis of the Protein–Protein Interaction (PPI) Network

The STRING online database (http://string-db.org (accessed on 13 December 2022), Version: 11.5) was used to construct the PPI network [64], and an interaction score > 0.4 was considered statistically significant. Analysis of functional interactions between proteins helped us to better understand the mechanisms of DNA methylation during testicular maturation. Plug-in MCODE and cytoHubba within Cytoscape [65] were used in turn to cluster the resulting network and identify hub genes of the network. The selection standards of the MOCODE were as follows: cutoff value = 2, node score cutoff value = 0.2, K-Core = 2, and maximum depth = 100. The maximal clique centrality (MCC) method in plug-in cytoHubba was used to screen hub genes [66].

### 4.8. Quantitative Real-Time PCR (qPCR)

qPCR was performed as described in our previous studies [19,31]. Total RNA was extracted using the TRIzol reagent (Invitrogen, Carlsbad, CA, USA), according to the protocols provided by the manufacturers. The primers were designed using Oligo 7.0 software (Integrated DNA Technologies, Coralville, IA, USA) (Appendix A) and synthesized by Sango Bio. (Shanghai, China). The cDNA of each sample was prepared using a cDNA synthesis kit (TaKaRa, Beijing, China). Qpcr was performed using a Thermo Scientific PIKO REAL 96 real-time PCR system, according to the SYBR Green kit instructions (TaKaRa, Beijing, China). *GAPDH* was used as an internal control, and each experiment was repeated three times. The gene expression level was normalized to *GAPDH* using the 2^−∆∆Ct^ method.

### 4.9. Sertoli Cell Culture and Zebularine Treatment

The immature porcine Sertoli cells were isolated from Shaziling piglets using the two-step enzymic method and the cell purity was more than 95%. Sertoli cells were cultured in Dulbecco’s modified Eagle medium/Nutrient Mixture F-12 (DMEM/F-12, Gibco, Grand Island, NY, USA), supplemented with 10% fetal bovine serum (Gibco, Grand Island, NY, USA) and 5% CO_2_, at 34 °C. Cells were seeded in a 6-well culture plate, at a density of 1 × 10^5^ cells/well. To study the effect of zebularine on gene expression, Sertoli cells were cultured in DMEM and treated with 15 μM zebularine (ApexBIO, Houston, TX, USA), for 72 h. Dimethyl sulfoxide (DMSO, Solarbio, Beijing, China) treatment was performed as a negative control (NC). Total RNA and DNA was extracted for further analysis.

### 4.10. Bisulfite Sequencing PCR (BSP)

BSP was used to measure the DNA methylation levels in promoter of *IFITM1* and exon of *FZD7.* DNA used for BSP was extracted by the animal genomic DNA kit (Tiangen, Beijing, China), according to the protocols provided by the manufacturers. The genomic DNA from zebularine treatment assay were treated with bisulfite. The processed DNA was used for touchdown PCR (ABI, Waltham, MA, USA). BSP primers were synthesized by Sango Bio. (Shanghai, China) and designed with the Primer Premier 5. Both BSP primer and target clone sequences are listed in the Appendix A. The PCR products were cloned into the pMD18-T vector (TaKaRa, Nagaokashi, Japan) and sequenced by Sango Bio. (Shanghai, China). The methylation levels were visualized by the CyMATE software (https://cymate.org/, 13 December 2022, Vienna, Austria, Version: 1.0).

### 4.11. RNA Interference (RNAi) of FZD7 and Transfection

Three *RNAi* oligonucleotides (si-*FZD7*-1, -2, -3) were designed and synthetized by RiboBio (Guangzhou, China), and the sequence of each siRNA are listed in the Appendix A. The Sertoli cells were seeded in a 6-well culture plate, at a density of 1 × 10^6^ cells/well and cultured with 2 mL medium. For the cell transfection, a total of 100 uM (final concentration: 50 nM in the cells) of siRNA or siRNA NC (si-NC) (RiboBio, China) was diluted with 120 μL of serum-free Opti-MEM (Thermo Fisher Scientific Inc., Grand Island, NY, USA), and incubated at 25 °C for 5 min. Next, 12 μL Lipofectamine^TM^2000 (Invitrogen, Carlsbad, CA, USA) was also diluted with mixed serum-free Opti-MEM (Gibco, Grand Island, NY, USA), and incubated at 25 °C for 30 min. Finally, the mixtures were added to each well when the cells reached approximately 80% confluence. After cultivation for 5 h, at 34 °C, with 5% CO_2_, complete medium was used for cultivation.

### 4.12. Immunofluorescence Staining

The fresh samples of testes from newborn Shaziling boars were fixed in 4% of formaldehyde for 72 h. Next, testes sections were made using routine histological methods. Slides were incubated for two hours, at 37 °C, with anti-SOX9 (1:3000, Proteintech Group, Chicago, IL, USA, 67439-1-Ig) or/and anti-FZD7 (1:1000, Bioss, Beijing, China, bs-5125R). After incubating with secondary antibodies for 1 h, at 37 °C, sections were soaked three times in phosphate-buffered saline (PBS) for 5 min each. DAPI (50 ng/mL, Sigma, Louis, MO, USA) was dripped onto testes slices and incubated at 26 °C, for 5 min, and then washed three times with PBS. All images were shot using a confocal laser scanning microscope (Zeiss LSM 510 META, GER).

### 4.13. Cell Cycle Assay

A cell cycle testing kit (Beijing 4A Biotech, Beijing, China) was used to analyze the cell cycle. After transfection for 24 h, Sertoli cells were soaked twice with PBS and collected in a 1.5 mL centrifuge tube. After incubating in 70% (*v*/*v*) ethanol, at −4 °C, for 12 h, cells were treated with propidium iodide (PI) solution (50 mg/mL) for 30 min, at 37 °C. Next, the prepared cell suspension was analyzed using an FACSCanto II Flow Cytometer (Becton Dickinson, Franklin Lakes, NJ, USA).

### 4.14. Cell Proliferation Assay

The immature Sertoli cells were seeded in 96-well plate, at a density of 1 × 10^3^ cells/well and transfected with si-*FZD7*-1 and si-NC. For the cell counting kit-8 (CCK-8) (Multiscience, Hangzhou, Zhejiang China) assay, after transfection for 24, 48, and 72 h, CCK-8 reagent was added to each well, and they were then incubated at 37 °C, for four hours. The absorbance was measured using the enzyme-linked immunosorbent assay (ELISA) plate reader (Thermo Fisher Scientific, Waltham, MA, USA), at 450 nm.

### 4.15. Cell Apoptosis Assay

After transfection for 24 h, cells were collected in a 1.5-mL centrifuge tube. Next, cells were washed three times and double-stained using an Annexin V-FITC apoptosis detection kit (Beijing 4A Biotech, Beijing, China). The cell apoptosis rates were analyzed using the FACSCanto II Flow Cytometer (Becton Dickinson, Franklin Lakes, NJ, USA). Percentages of early apoptosis and late apoptosis cells were counted and used as the cell apoptosis rate.

### 4.16. Western Blotting

After transfection for 48 h, the total cellular protein was extracted using the radio immunoprecipitation assay (RIPA) lysis buffer (Beyotime, Shanghai, China), and the protein concentrations were measured using the bicinchoninic acid protein assay kit (BCA) (Beyotime, Shanghai, China). The boiled protein samples were electrophoresed on 10% SDS-polyacrylamide gels, and then transferred onto a PVDF membrane (Beyotime, Shanghai, China). The membrane containing protein fractions was blocked with QuickBlock^TM^ Blocking Buffer (Beyotime, Shanghai, China) for 30 min and incubated with primary antibodies for 12 h, at 4 °C, including FZD7; β-Catenin (1:2000, Huabio, Hangzhou, Zhejiang, China, 0407-16); BAX (1:2000, Proteintech Group, Chicago, IL, USA, 50599-2-Ig); Bcl2 (1:1000, Bioss, Beijing, China, bsm-33047M); Caspace-3 (1:1000, Cell Signaling Technology, Danvers, MA, USA, 14220); and β-actin (1:5000, Proteintech Group, Chicago, IL, USA, 66009-1-Ig). After washing, the membrane was incubated with secondary antibodies for 2 h, at 26 °C. Protein bands were visualized using an ECL advanced western blotting detection kit (Beyotime, Shanghai, China). β-actin served as the loading control.

### 4.17. Measurement of the Intracellular Ca^2+^ Concentration

After transfection for 24 h, Sertoli cells were counted and collected in a 1.5-mL centrifuge tube. Cells were transferred to a digestion tube filled with 50 mL Teflon. Next, 9 mL nitric acid (concentration: 68%) and 1 mL perchloric acid (concentration: 70%) were added to the digestion tubes, which were sealed overnight, at 25 °C. Next, digestion tubes were placed on an electric hot plate until complete digestion, then the digestion solution was transferred to a 50 mL volumetric flask and made up to volume with nitric acid solution (concentration: 1%). In addition, the reagent blank was constructed by repeating the above steps, without cells. Finally, the Ca^2+^ levels of mixed digestion solutions were analyzed using ICP-MS (Agilent 7800, Santa Clara, CA, USA).

### 4.18. Adenosine Triphosphate (ATP) Assay

After transfection for 24 h, cells were collected in a 1.5-mL centrifuge tube. The ATP concentration were measured using the ATP assay kit (Beyotime, Shanghai, China), according to the manufacturer’s protocols. The relative light unit (RLU) was measured using Chemiluminescence Apparatus (Thermo Fisher Scientific Inc., Waltham, MA, USA).

### 4.19. Prostaglandin E_2_ (PGE_2_) Assay

The PGE_2_ level was examined using a porcine PGE_2_ kit (Shanghai YuanjuBio, China) on an enzyme-linked immunosorbent assay (ELISA) plate reader, and Sertoli cells were seeded in a 6-well culture plate. After transfection for 24, 48, and 72 h, cell supernatants were collected by centrifugation at 3000 rpm, for 10 min. The absorbance was measured using an ELISA plate reader (Molecular Devices, Silicon Valley, CA, USA) at 450 nm, and then the activin A concentration was calculated.

### 4.20. Apoptosis-Medium Treatment

To demonstrate Sertoli cells’ repopulation by cell apoptosis, apoptosis medium collected during cell apoptosis was used to treat Sertoli cells. Firstly, cells were seeded in a 6-well plate, and were cultured using DMEM, without fetal bovine serum, after the cells reached approximately an 60% confluence. Next, cell supernatants were immediately collected by centrifugation at 3000 rpm, for 10 min, after suspended cells began to present. Those supernatants were named apoptosis medium in this present study. Furthermore, Sertoli cells were cultured using a mixed medium: 10% fetal bovine serum, and an equal volume of apoptosis medium and DMEM. In addition, Sertoli cells cultured using DMEM, with 10% fetal bovine serum, were identified as NC. Finally, ATP assay and qPCR for genes associated with proliferation were performed after culturing for 48 h, and cell proliferation assay was performed as mentioned above.

### 4.21. Statistical Analysis

In the present study, each treatment group contained three independent biological replicates. Means between experimental and control groups were compared using a *t*-test at a single time point. The test data were analyzed using SAS^®^ OnDemand for Academics (https://odamid-apse1.oda.sas.com/ (accessed on 30 December 2022)). *p* < 0.05 and *p* < 0.01 were considered as statistically significant.

## Figures and Tables

**Figure 1 ijms-24-06179-f001:**
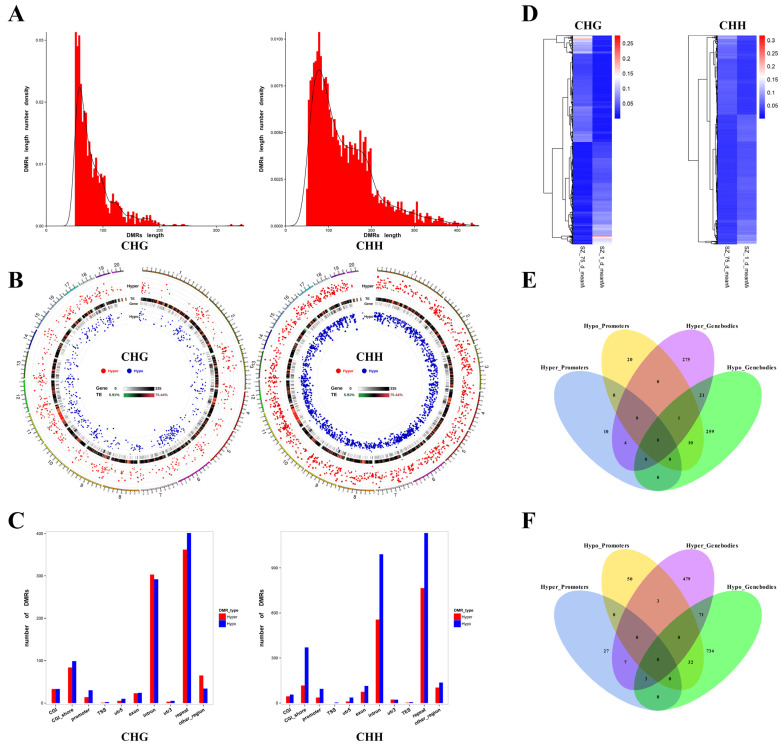
Characteristics of DMRs and DMGs. (**A**) The frequency distribution of two types of DMR count, by length. (**B**) Circos plots of the distribution and significance of two types of DMRs. The outer ring represents the chromosome of the reference genome, and the chromosome information corresponding to the number is listed in Appendix A. Circos plots were generated according to the length of the chromosome, from long to short: hyper—the distribution of the hypermethylated DMRs, in which the differential methylation level is positively correlated with the distance from the center of the circle; TE—the proportion of repetitive sequences, which turn from green to red, indicates the proportion from low to high; gene—the density of genes in each bin, in which turning from white to black indicates the density becoming larger; and hypo—the distribution of hypomethylated DMRs, in which the differential methylation level is negatively correlated with the distance from the center of the circle. (**C**) Distribution of two types of DMR among genomic elements. (**D**) Heat maps of DMR methylation levels in CHG and CHH contexts between two developmental stages. (**E**,**F**) Show the composition of DMGs associated with CHG and CHH contexts, respectively: Hyper_Promoters—hypermethylated DMGs associated with promoters; Hypo_Promoters—hypomethylated DMGs associated with promoters; Hyper_Genebodies—hypermethylated DMGs associated with gene bodies; and Hypo_Genebodies—hypomethylated DMGs associated with gene bodies.

**Figure 2 ijms-24-06179-f002:**
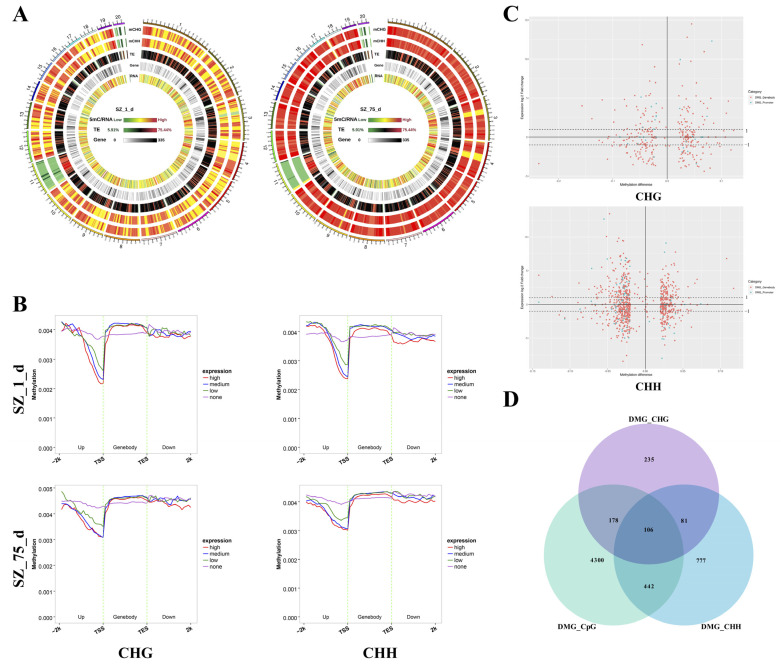
Relationship between non-CpG methylation and gene expression. (**A**) DNA methylation features of two developmental stages at the chromosome level. The outer ring represents the chromosome of the reference genome, and the chromosome information corresponding to the number listed in Appendix A. mCHG—methylation density in the CHG context; mCHH—methylation density in the CHH context. The 5mC corresponding colors represent methylation density: green, low, and red, high. TE—proportion of repetitive sequences: green, low, and red, high. Gene—the density of genes in each bin: white, low, and black, high. (**B**) Average methylation levels of two types of contexts across and around gene bodies of genes with different expression levels, at two developmental stages. Genes were classified into four groups according to their expression level (no expression, low expression, medium expression, and high expression), by RNA-seq. (**C**) Differential methylation levels and corresponding differential expression levels of DMGs in two types of contexts. (**D**) DMGs of three types of contexts.

**Figure 3 ijms-24-06179-f003:**
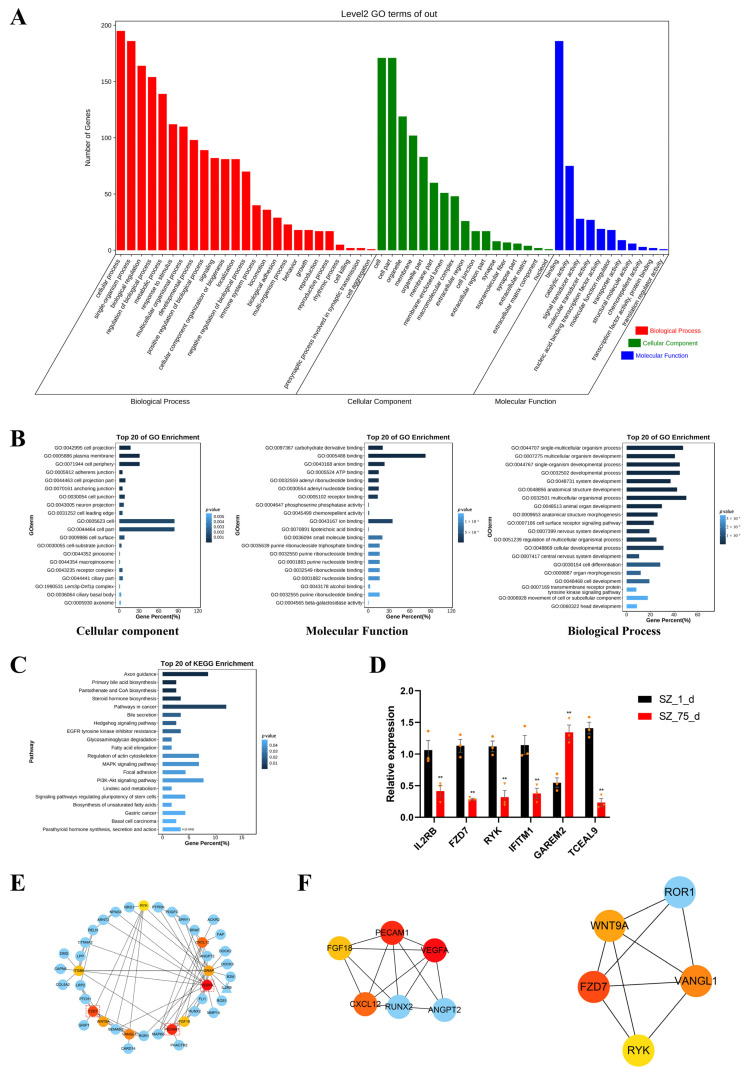
Non-CpG methylation regulated testicular development through multiple pathways. (**A**) GO annotation was performed at Level 2 for three main categories. (**B**) Top 20 GO terms for three main categories. (**C**) Top 20 KEGG pathways. (**D**) The relative mRNA levels of BPGs and PNGs at two developmental stages. The GAPDH gene was used as the internal control. Orange triangles and circles represented the numerical distribution of three biological repeats in each group. Data were presented as the mean ± SEM. ** *p* < 0.01. (**E**) Identification of hub genes from BPGs and PNGs by STRING and Cytoscape. Top ten hub genes are marked with warm colors, and color changes show the degree of genes calculated by the CytoHubba plug-in. Ellipse—BPGs; and triangle—PNGs. (**F**) Two functional modules filtered out by the MOCODE plug-in.

**Figure 4 ijms-24-06179-f004:**
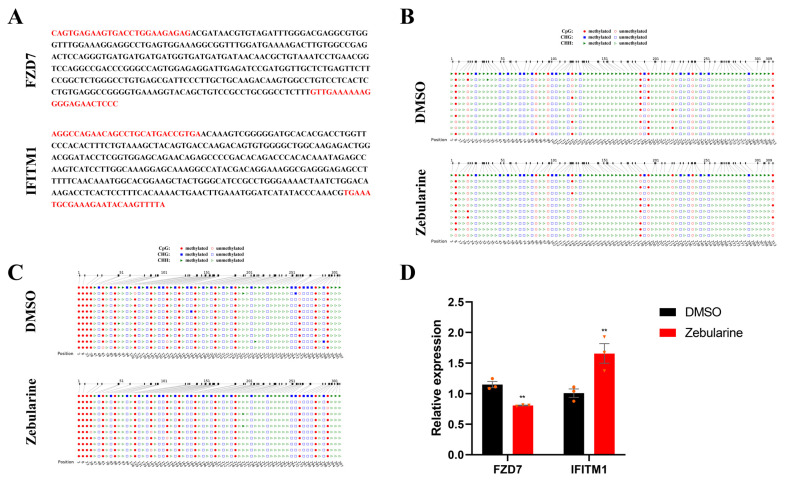
Results of zebularine treatment. (**A**) Target amplified sequences of *FZD7* and *IFITM1* for BSP. Bases marked in red are the primer positions. (**B**,**C**), Demethylation sites of *FZD7* and *IFITM1*, by treating with zebularine. (**D**) The relative mRNA levels of *FZD7* and *IFITM1* between two treatments. Orange triangles and circles represented the numerical distribution of three biological repeats in each group. Data are presented as the mean ± SEM. ** *p* < 0.01.

**Figure 5 ijms-24-06179-f005:**
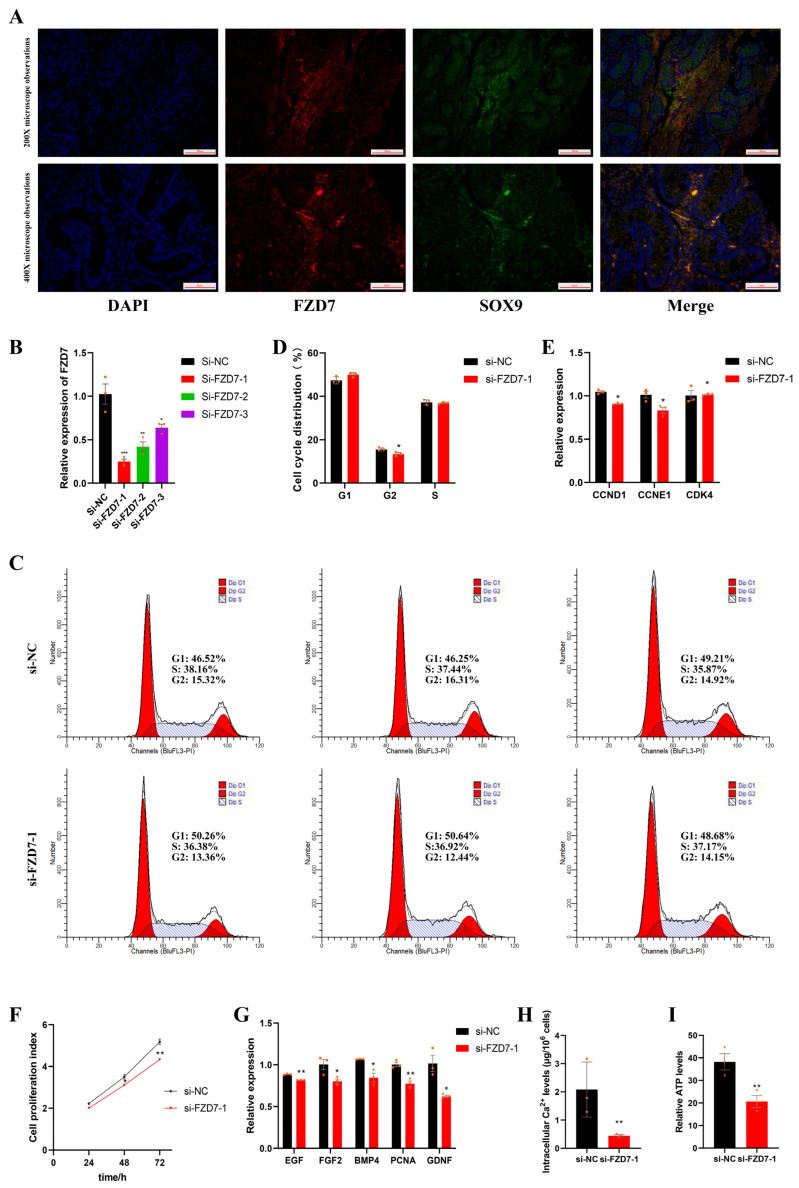
*FZD7* promotes the proliferation of immature porcine Sertoli cells. (**A**) The location of *FZD7* in porcine testicular tissue. Testicular tissues of newborn pigs were used in this experiment. *FZD7* is marked in red and SOX9 is marked in green. (**B**) Sertoli cells were treated with RNAi for *FZD7* (si-*FZD7*-1, -2, and -3). The inhibition efficacy was evaluated using the qPCR assay. si-NC—negative control of RNAi. (**C**) The CCK-8 assay was used to measure the cell proliferation index. (**D**) Relative mRNA expression of the key genes of cell proliferation. (**E**,**F**) show the cell cycle was analyzed using a FACSCanto II flow cytometer, and the G1, S, and G2 phases of the cell cycle were counted in cells transfected with si-*FZD7*-1. (**G**) Relative mRNA expression of the key genes of cell cycles. (**H**) Intracellular Ca^2+^ levels were measured using ICP-MS. (**I**) The relative ATP level was measured using an ATP assay kit. Orange triangles and circles represented the numerical distribution of three biological repeats in each group. Data were presented as the mean ± SEM. * *p* < 0.05, ** *p* < 0.01 and *** *p* < 0.001.

**Figure 6 ijms-24-06179-f006:**
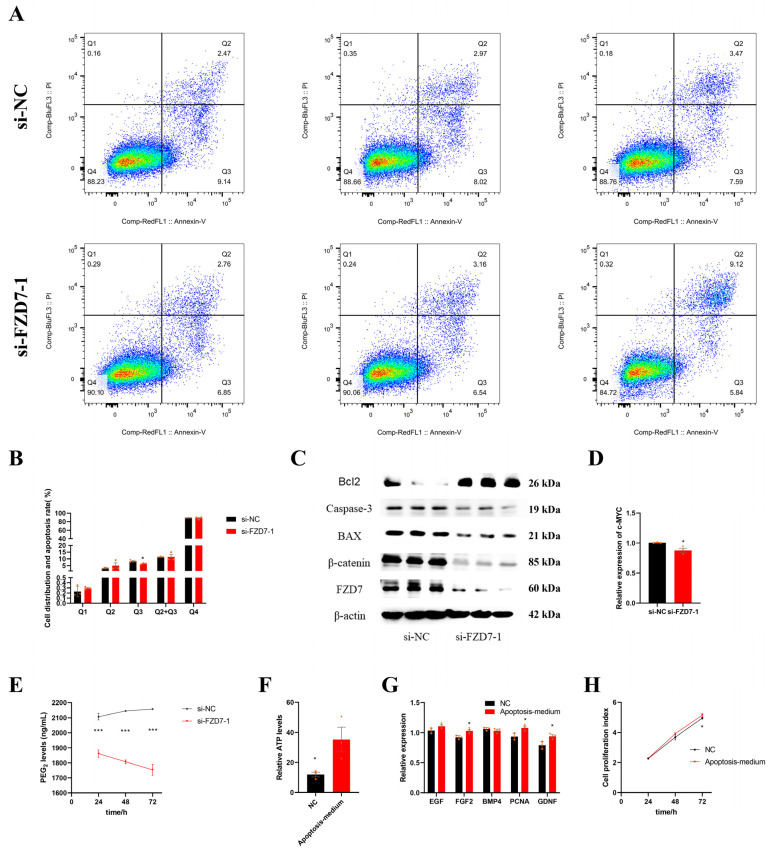
Sertoli cells’ repopulation by the early apoptosis. (**A**,**B**) Cell apoptosis phase distributions were detected using Annexin V-FITC/PI staining assay. Q1—the percentage of necrotic cells; Q2—the percentage of late apoptotic cells; Q3—the percentage of early apoptotic cells; Q4— non-apoptotic cells. (**C**) Protein expression of cell apoptosis marker genes, and those proteins were determined using western blot analysis. The β-actin gene was used as the internal control. (**D**) Relative mRNA expression levels of *c-MYC*. (**E**) The PGE2 level was measured using a porcine PGE2 ELISA kit. (**F**) The relative ATP level for apoptosis-medium treatment. (**G**) Relative mRNA expression of the key genes of cell proliferation for apoptosis-medium treatment. (**H**) Cell proliferation index for apoptosis-medium treatment. Orange triangles and circles represented the numerical distribution of three biological repeats in each group. Data are presented as the mean ± SEM. * *p* < 0.05. and *** *p* < 0.001.

**Figure 7 ijms-24-06179-f007:**
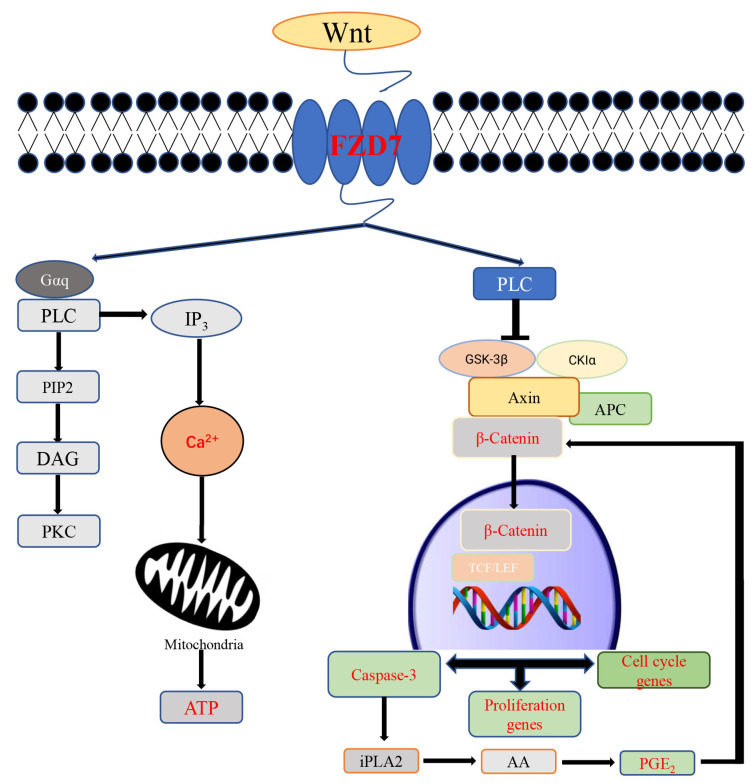
Model of the main investigations in this study.

**Table 1 ijms-24-06179-t001:** Ranking of the top 10 hub genes.

Rank	Gene Symbol	Score
1	*VEGFA*	36
2	*PECAM1*	21
3	*FZD7*	17
4	*CXCL12*	16
5	*VANGL1*	15
6	*WNT9A*	14
6	*GRAP*	14
8	*FGF18*	12
8	*ITGB6*	12
10	*RYK*	10

**Table 2 ijms-24-06179-t002:** Percentage of methylated cytosines in the three types of contexts.

Genes	Zebularine-Treated Group			DMSO-Treated Group		
	CpG	CHG	CHH	CpG	CHG	CHH
*FZD7*	95.62	0	0.71	96.88	1.36	1.07
*IFITM1*	25	0	0	30	0	0.18

## Data Availability

The datasets and results presented in this study can be found in online repositories. The names of the repository/repositories and accession number(s) can be found in the article/Appendix A.

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
