# Peer review of "FZD7, Regulated by Non-CpG Methylation, Plays an Important Role in Immature Porcine Sertoli Cell Proliferation"

_ijms, 2023, doi:10.3390/ijms24076179_

Round 1
Reviewer 1 Report
The literature article by Anqi et al. describes studies FZD7 regulated by non-CpG methylation plays an important role in immature porcine Sertoli cell proliferation . And exploring the regulatory mechanisms of non-CpG methylation in testicular development are important issues in development and health.
General comments:
The writing is not very concise and there are many grammatical errors and typos, which makes the manuscript hard to follow in places. Some sentences are redundant in introduction cites, it has to be more concise and accurately reflect the points discussed. The authors should revise the language to improve readability. Like line 33-35, I can't find the this conclusion in cited paper.
Results are correct and complete. However, the Figure 1 and 2 are not clear. Please update your Figure 1 and 2 quality. And also label the genes (for example VEGFA, PECAM1, and FZD7) that you are interested in circos diagrams to show the difference.
Figure 3 B, C and E are also not clear.
Figure 5 A I can't find Sertoli cell signal in your data. It seems like autofluorescence in the tubules.
Figure 5 E Please check your location of statistic label.
The presentation of the results is not adequate since the authors report the data do not follow a normal distribution. Please check your flow cytometry figures with your data presentation carefully.
Author Response
Dear Editors and Reviewers,
Thanks for your comments concerning our manuscript entitle “FZD7 regulated by non-CpG methylation plays an important role in immature porcine Sertoli cell proliferation” (ijms-2249256). Those comments are all valuable and very helpful for revising and improving our manuscript. We have made correction according to the comments of editors and reviewers. The main corrections in the paper and the responds to the reviewer’s comments are shown in PDF

Reviewer 2 Report
This is a very interesting and comprehensive research on the role of FZD7 in the proliferation of immature Sertoli cells. The Results are clearly presented and the Discussion focus on the relevance of the Results obtained. The manuscript is well written, but it contains several typographic error, and some grammatical errors that must be corrected.
My only concern refers to the first paragraph of the Introduction referring to the physiological role of Sertoli cells. The information given is too vague and the text contain several grammatical errors that provide a puzzling message. For instance, in lines 31-32 the sentence “Sertoli cells are the largest and heterogeneously shaped cells, and located in the germinal epithelium in the seminiferous tubules” is grammatically incorrect. On the other hand, in the sentence of lines 35-36. “Sertoli cells can also promote spermatogenesis by secreting hormones” is very ambiguous. Authors must specify the hormones secreted by these cells. The sentence of lines 39-41 must also be improved.
Author Response
Dear Editors and Reviewers,
Thanks for your comments concerning our manuscript entitle “FZD7 regulated by non-CpG methylation plays an important role in immature porcine Sertoli cell proliferation” (ijms-2249256). Those comments are all valuable and very helpful for revising and improving our manuscript. We have made correction according to the comments of editors and reviewers. The main corrections in the paper and the responds to the reviewer’s comments are as following:
Responds to the reviewer’s comments:
Reviewer:2
This is a very interesting and comprehensive research on the role of FZD7 in the proliferation of immature Sertoli cells. The Results are clearly presented and the Discussion focus on the relevance of the Results obtained. The manuscript is well written, but it contains several typographic error, and some grammatical errors that must be corrected.
My only concern refers to the first paragraph of the Introduction referring to the physiological role of Sertoli cells. The information given is too vague and the text contain several grammatical errors that provide a puzzling message. For instance, in lines 31-32 the sentence “Sertoli cells are the largest and heterogeneously shaped cells, and located in the germinal epithelium in the seminiferous tubules” is grammatically incorrect. On the other hand, in the sentence of lines 35-36. “Sertoli cells can also promote spermatogenesis by secreting hormones” is very ambiguous. Authors must specify the hormones secreted by these cells. The sentence of lines 39-41 must also be improved.
Thanks your valuable comments. We have carried out extensive English revisions and updated all figures, please check our manuscript.
Round 2
Reviewer 1 Report
Author have addressed all my concern.